# OpenReview forum: "Leak@$k$: Unlearning Does Not Make LLMs Forget Under Probabilistic Decoding"
_ICML.cc/2026/Conference — ICML 2026 regular_

### Official Review · Reviewer_V3Uw · 2026-03-09

**Soundness:** 3
**Presentation:** 3
**Significance:** 2
**Originality:** 3
**Overall Recommendation:** 6
**Confidence:** 4

**Summary:**

This paper states that almost all existing unlearning methods fail to achieve true forgetting in LLM. Empirical results are presented to justify the argument. A new evaluation metric is proposed to measure the likelihood of forgotten knowledge reappearing. Experiments are conducted on benchmarks including TOFU, MUSE, and WMDP.

**Compliance With Llm Reviewing Policy:**

Affirmed.

**Final Justification:**

The proposed new evaluation looks novel to me. Experiments are conducted on various benchmarks to justify the usefulness of the evaluation. The major concerns have been addressed in the rebuttal. The overall quality is over the bar in my pool.

**Key Questions For Authors:**

Please refer to the weakness.

**Limitations:**

Any evaluation has its limitations. A paragraph of limitations can make the paper complete.

**Strengths And Weaknesses:**

Strengths:

1. The paper points out the issues of existing unlearning evaluation. The proposed evaluation could benefit the unlearning community. It is urgent to develop new evaluations for machine unlearning, especially for LLM unlearning.
2. The paper studies knowledge resurface, which is an interesting phenomenon in LLM.
3. Worst estimation is an intuitive evaluation and provide statistical measurement of data leakage.

Weakness:

Soundness:
1. The value of k is vital in the evaluation. The authors have discussed the values of k. However, it is not very conclusive what the value of k should be. It could depend on the benchmarks and models. Such a hyperparameter is not practical and will pose further open arguments regarding evaluation if used, e.g., it is challenging to reach an agreement in the community. Thus, I doubt this evaluation will be used widely in both research and practice.

2. While the motivation is knowledge resurfacing, more attack-based unlearning methods [1] should be adopted in the experiment.

3. Cross-validation between leak@k and other evaluation such as MIA, should be conducted, if the evluation align with other evaluations in what cases.

[1] Unlearning Sensitive Information in Multimodal LLMs: Benchmark and Attack-Defense Evaluation

Significance:

4. I am wondering if such an evaluation can be used for other models, such as diffusion models [2, 3]. If the claim is only for LLM, the impact will be a little limited.

[2] Sculpting Memory: Multi-Concept Forgetting in Diffusion Models via Dynamic Mask and Concept-Aware Optimization.
[3] Defensive Unlearning with Adversarial Training for Robust Concept Erasure in Diffusion Models

Originality:

5. On the high level, a similar idea, also based on sampling, is proposed in [4].

[4] A Fully Probabilistic Perspective on Large Language Model Unlearning: Evaluation and Optimization

---

> ### Author Rebuttal · Authors · 2026-03-31
>
> > The value of k is vital in the evaluation. The authors have discussed the values of k. However, it is not very conclusive what the value of k should be. It could depend on the benchmarks and models. Such a hyperparameter is not practical and will pose further open arguments regarding evaluation if used, e.g., it is challenging to reach an agreement in the community. Thus, I doubt this evaluation will be used widely in both research and practice.
>
> **Response:** We disagree with the reviewer that there should be a fixed $k$ for the evaluation. In our setting, $k$ represents the **capability** of user to extract the sensitive information. Fixing a single $k$ does not show how risk scales with repeated sampling.
>
> Prior work in the pass@k literature explicitly treats $k$ as an attempt/sampling budget, not a fixed constant. For example, [1] evaluates with up to 100 samples per problem and show that performance depends strongly on $k$ and the sampling strategy. Subsequent work consistently reports multiple operating points (e.g., pass@1, pass@5, pass@10), rather than using a single value [2]. This reflects a well-established view: $k$ is application-dependent, not a hyperparameter that requires an universal agreement.
>
> Thus, the flexibility in $k$ is not a limitation but a **necessary feature** for meaningful evaluation under realistic usage.
>
>
> > While the motivation is knowledge resurfacing, more attack-based unlearning methods [1*] should be adopted in the experiment. Cross-validation between leak@k and other evaluation such as MIA, should be conducted, if the evluation align with other evaluations in what cases.
> [1*] Unlearning Sensitive Information in Multimodal LLMs: Benchmark and Attack-Defense Evaluation
>
> **Response:** We thank the reviewer for highlighting the importance of incorporating additional attack-based unlearning evaluations. We therefore conduct cross-validation between MIA and Leak@k on the TOFU benchmark. As shown in the table, both metrics exhibit **consistent method ranking** (Retrain < GradDiff < NPO < Original) and coherent trends across all settings, indicating strong alignment in measuring residual memorization. This suggests that Leak@k and MIA capture the same underlying unlearning effectiveness, while Leak@k **provides evaluation from an adversarial perspective**, capturing worst-case information leakage under probabilistic decoding as the sampling budget increases.
>
> | Method | MIA (~0.5) | Leak@1 (↓) | Leak@2 (↓)| Leak@4 (↓)| Leak@8 (↓)| Leak@16 (↓)| Leak@32 (↓)| Leak@64 (↓)| Leak@128 (↓)|
> | ------ | -------------- | ----------------- | ------ | ------ | ------ | ------- | ------- | ------- | -------- |
> | Retrain  | 0.4767 | 16.9 | 23.6 | 30.6 | 37.6 | 44.1 | 50.1 | 55.7 | 61.0 |
> | GradDiff | 0.9232 | 26.1 | 36.7 | 47.4 | 57.3 | 65.8 | 72.8 | 78.5 | 83.0 |
> | NPO      | 0.7491 | 20.4 | 29.2 | 38.4 | 47.3 | 55.4 | 62.5 | 68.7 | 74.3 |
> | Original | 0.9980 | 28.7 | 40.5 | 52.4 | 63.1 | 72.1 | 79.4 | 84.9 | 89.1 |
>
> Table. Cross-validation between MIA and Leak@k on the TOFU benchmark
>
> > I am wondering if such an evaluation can be used for other models, such as diffusion models [2*, 3*]. If the claim is only for LLM, the impact will be a little limited.
> [2*] Sculpting Memory: Multi-Concept Forgetting in Diffusion Models via Dynamic Mask and Concept-Aware Optimization.
> [3*] Defensive Unlearning with Adversarial Training for Robust Concept Erasure in Diffusion Models
>
> **Response:** We thank the reviewer for this insightful suggestion. Our proposed Leak@k framework is not intrinsically limited to LLMs, but rather operates at the level of **stochastic generative sampling under probabilistic decoding**. In diffusion models [2*,3*], this corresponds naturally to sampling multiple outputs per prompt across different noise seeds and prompt perturbations. Leak@k can thus be directly instantiated as a worst-case audit over k generated images, measuring whether erased concepts resurface under repeated sampling rather than a single deterministic generation. Therefore, we view our framework as a **general adversarial evaluation principle for generative unlearning across modalities**.
>
> > On the high level, a similar idea, also based on sampling, is proposed in [4*].
>
> [4*] A Fully Probabilistic Perspective on Large Language Model Unlearning: Evaluation and Optimization
>
> **Response:** We thank the reviewer for pointing out the close connection to prior probabilistic unlearning work [4*]. FPE [4*] and related methods provide a principled probabilistic evaluation framework, while our approach Leak@k complements them by focusing on adversarial multi-sample leakage amplification, which is particularly relevant for security-sensitive unlearning scenarios. We view these directions as orthogonal and complementary: probabilistic averaging captures expected risk, whereas Leak@k captures worst-case retrievability under repeated sampling.

---

> > ### Author Rebuttal · Reviewer_V3Uw · 2026-04-03
> >
> > My major concerns have been addressed. Including the discussion on diffusion models in the revision would increase the impact of the proposed method. My current recommendation score has already reflected the overall contribution.

---

> > > ### Author Response · Authors · 2026-04-04
> > >
> > > Thank you very much for your follow-up and for confirming that our rebuttal has fully addressed your concerns. We sincerely appreciate your thoughtful review and the constructive suggestion to include a discussion on diffusion models, which we agree would further enhance the impact of the paper.
> > >
> > > If possible, we would greatly appreciate it if you could consider updating your score or confidence to reflect your latest assessment.

---

### Official Review · Reviewer_3cko · 2026-03-11

**Soundness:** 2
**Presentation:** 2
**Significance:** 2
**Originality:** 2
**Overall Recommendation:** 4
**Confidence:** 2

**Summary:**

This work critiques the use of greedy decoding to evaluate the existing unlearning methods. They make an argument that the information is never truly unlearned and can be extracted by using non-deterministic sampling methods like top-k or nucleus sampling. Furthermore, the authors propose their own unlearning method to address this issue.

**Compliance With Llm Reviewing Policy:**

Affirmed.

**Final Justification:**

Authors addressed my main concerns in the rebuttal,

**Key Questions For Authors:**

- Can the authors clarify what the amount of paraphrase leakage was while they were evaluating the models?

**Limitations:**

yes

**Strengths And Weaknesses:**

### Strengths
- The claims in this paper are well supported through different experiments, math, and current literature.
- Evaluating unlearning is an active area of research and a very important one. This work makes an effort to contribute in this direction.
- The paper is well-written and easy to follow for the readers.

### Weaknesses
- My main critique of this work is the novelty of the entire idea. The fact that information that is not recovered (or leaked) under greedy decoding may be leaked under non-deterministic sampling (like top-k or nucleus sampling) has been well-studied before (for example, https://arxiv.org/abs/2410.19482, and a couple more). The authors neither cite nor acknowledge any of these prior works in their study, which creates a wrong illusion that this whole idea is novel.
- Some of the results indicate that we need to set the value of k to be very high (128, as reported in the paper) to break many of the unlearning methods. This makes me question the scalability of this approach. Having a much lower number would make these results more interesting and convincing.
- Even though the work measures the impact of unlearning on model utility (which is nice), I'd like to see how it affects capabilities like reasoning and fluency.

I agree that this is an important direction of research, but addressing the first two weaknesses would make this work more impactful.

---

> ### Author Rebuttal · Authors · 2026-03-31
>
> > My main critique of this work is the novelty of the entire idea. The fact that information that is not recovered (or leaked) under greedy decoding may be leaked under non-deterministic sampling (like top-k or nucleus sampling) has been well-studied before (for example, https://arxiv.org/abs/2410.19482, and a couple more). The authors neither cite nor acknowledge any of these prior works in their study, which creates a wrong illusion that this whole idea is novel.
>
> **Response:** We agree that [7] is a relevant work and will cite it in the final version of our paper. However, their setting is probabilistic extraction of a known target sequence from **pretrained** models, typically using a prefix–suffix formulation and often computable from token probabilities in a **single query**. Our setting is different: we evaluate **post-unlearning** residual leakage on unlearning benchmarks (TOFU, MUSE, WMDP), where leakage is defined by benchmark-level semantic/task criteria, not only exact target-string extraction.
>
> [7] Hayes et al., Measuring memorization in language models via probabilistic extraction, 2025.
>
> > Some of the results indicate that we need to set the value of k to be very high (128, as reported in the paper) to break many of the unlearning methods. This makes me question the scalability of this approach. Having a much lower number would make these results more interesting and convincing.
>
> **Response:** We thank the reviewer for the insight regarding scalability. While k=128 represents a rigorous stress test for the most robust models, most unlearning methods already exhibit significant, actionable leakage at much lower values (e.g., k=8). The parameter k essentially defines the **adversarial budget**; by evaluating across a range, we capture the full spectrum of risk from casual queries to determined extraction attempts. Crucially, as detailed in Table A4, a complete evaluation up to k=128 requires only approximately 1 hour, confirming that our framework remains **computationally efficient** and **highly scalable for practical benchmarking.**
>
> > Even though the work measures the impact of unlearning on model utility (which is nice), I'd like to see how it affects capabilities like reasoning and fluency.
>
> **Response:** We thank the reviewer for this suggestion. We evaluate **fluency** (WikiText PPL) and **reasoning** (ARC, GSM8K), and reorganize results as follows.
>
> | Metric | RULE (NPO) | RULE (GradDiff) | NPO | GradDiff | Original | Retrain |
> |--------|------------|----------------|-----|----------|----------|---------|
> | WikiText PPL ↓ | 31.93 | 24.71 | 26.47 | 21.98 | 21.30 | 21.09 |
> | ARC Acc ↑ | 0.352 | 0.348 | 0.369 | 0.378 | 0.378 | 0.380 |
> | GSM8K Acc ↑ | 0.229 | 0.287 | 0.212 | 0.334 | 0.321 | 0.341 |
>
> While all unlearning methods incur a slight utility cost, our RULE framework demonstrates **minimal degradation compared to standard unlearning baselines**. This marginal decrease in fluency and reasoning is **a small and justifiable expense** for the significantly stronger unlearning performance and adversarial robustness that RULE provides.
>
> > Can the authors clarify what the amount of paraphrase leakage was while they were evaluating the models?
>
> **Response:** Our current evaluation does not explicitly isolate *paraphrase leakage*. The leak@k metric is computed using ROUGE-based overlap, which captures lexical and near-verbatim overlap but does not detect paraphrases. Also, we used accuracy-based metrics relies on exact matching.

---

> > ### Author Rebuttal · Reviewer_3cko · 2026-04-03
> >
> > Thanks for providing important clarifications. I'd like to increase my score.

---

> > > ### Author Response · Authors · 2026-04-04
> > >
> > > Thank you for reviewing our paper and for your kind acknowledgment. We truly appreciate your feedback and support, and are glad our clarifications were helpful.

---

### Official Review · Reviewer_z5Dj · 2026-03-13

**Soundness:** 3
**Presentation:** 2
**Significance:** 2
**Originality:** 1
**Overall Recommendation:** 4
**Confidence:** 4

**Summary:**

This paper is motivated by the observation that knowledge which appears "unlearnt" under greedy decoding my re-emerge under non-greedy settings, i.e., probabilistic decoding. The authors introduce the metric "Leak@k" which aims to assess the extent to which such knowledge is leaked as under probabilistic decoding methods. Specifically, `k` generations are sampled from a model and the `Leak@k` score is the probability that a generation leaks information from the forget set. The authors show that unlearning methods fail according to Leak@k under probabilistic decoding. Interestingly, even retrain baselines also fail to some extent.

**Compliance With Llm Reviewing Policy:**

Affirmed.

**Final Justification:**

Many of my concerns were addressed.

**Key Questions For Authors:**

- How does the estimator in Eq. 3 relate to the pass@k estimator derived by Chen et al. (2021)? Can you clarify the differences? If they are substantially similar, can you please cite Chen et al. properly?
- Across many of the heatmaps, the "retrain" model exhibits significant leakage. Have you thoroughly analyzed what is driving this leakage? If not, can you please do so and clarify. If the benchmark itself is contaminated, that would be good to know. If this is an artifact of using the worst-case leakage, that is also good to know.
- In Table A5, it looks like the average leakage is similar for greedy and probabilistic decoding, 19% and 20.5%, respectively (again, without CIs it is hard to interpret this). This suggests that the high leakage rates are concentrated in the worst-case version of the measure. How do you reconcile this with the paper's claim that unlearning methods fail under probabilistic decoding?
- Have you explored RULE on any of the other unlearning benchmarks such as MUSE-News? The TOFU benchmark you evaluate on is synthetic data, which is arguably the easiest unlearning setting.
- Have you explored whether RULE is scalable to larger datasets and models? 2hrs seems like a long time for a small dataset and model.
- Have you measured the uncertainty in the point estimates reported? Are the differences statistically significant? How much variation is there from random seed to random seed?

**Limitations:**

- The authors need to address the fact that the retrain model, which by definition should not exhibit leakage, is exhibiting material leakage across a variety of tests. If this is an artifact of the measure, researchers need to know that. Similarly, if there is a problem with the underlying benchmark, we need to know that.
- If RULE is indeed limited in applicability to smaller datasets and models, then the authors need to acknowledge that.
- The paper should specifically state the fact that the results are limited to models <10B scale and may not generalize

**Strengths And Weaknesses:**

### Strengths
- This is an important topic and the authors do a good job of motivating it. Modern sampling methods generally use probabilistic decoding setups. Thus, if unlearning methods are being applied with the hope that they are effective, yet the use probabilistic decoding immeadiately unwinds their efficacy, then that is important to know.
- The experiments are quite comprehensive: three of the main unlearning benchmarks, all of the classic unlearning methods and many of the recent methods
- The finding that across all of these methods, the Leak@k reaches surprisingly high levels, even for moderate k is interesting and important
- I like the RULE algorithm demonstrations which show that we can improve the robustness of unlearning methods relative to the Leak@k metric

### Weaknesses
- The observation that non-greedy decoding may lead to overly optimistic estimates of the efficacy of unlearning methods is not a new idea. Scholten et al. (2024) do the same, and introduce a variety of metrics which aim to capture the output distribution of a model. The paper acknoledges Scholten et al., but would benefit from a more candid discussion of the comparatively incremental nature of the contribution.
- The main contribution of the paper is the Leak@k metric. But this is similar, arguably identical, to pass@k (Chen et al., 2021). The paper mentions Chen et al. in passing but does not acknoledge that Chen et al. is effectively the predecessor to the Leak@k metric. Chen et al., effectively performed the same derivation without the continuous threshold.
- The empirical pattern that "leakage consistently increases with k" is entirely mechanical. I.e., the expected maximum of k iid draws from any non-degenerate probability distribution is monotonically non-decreasing with k. The authors should not focus on this pattern, which is mechanical, and instead focus on the level of the leakage.
- Different metrics are used from one benchmark to the next. This makes it difficult to compare across the benchmarks
- Explorations are limited to smaller models, on the order of 1-7B parameters. It is unclear whether the findings will generalize to larger models where unlearning dynamics can differ significantly.
- While I like the RULE approach, it is only applied to TOFU
- There are no estimates of uncertainty anywhere in the paper

---

> ### Author Rebuttal · Authors · 2026-03-31
>
> > The observation that non-greedy decoding ... a more candid discussion of the comparatively incremental nature of the contribution.
>
> **Response:**  While Scholten et al. (2024) adopts probabilistic evaluation, our contribution **differs in perspective, methodology and algorithm.**
>
> 1. **Evaluation perspective**. Leak@k focuses on **worst-case leakage**.
> 2. **Semantic fidelity.** Leak@k incorporates **semantic-level evaluation**
> 3. **Algorithm improvement**. In Leak@k, we introduce RULE to effectively mitigate leakage.
>
> > The main contribution of the paper is the Leak@k..to pass@k (Chen et al., 2021)...
>
> **Response:** We emphasize a key conceptual shift. Pass@k uses multiple samples to increase the chance of **task success**, whereas Leak@k uses the same multi-sample mechanism in the **opposite direction**.
>
>
> > The empirical pattern that "leakage consistently increases with k" is entirely mechanical ... and instead focus on the level of the leakage.
>
> **Response:** **Response:** Our contribution is not to this monotonicity, but to characterize **how leakage grows with $k$**, which directly relates to the goal of unlearning, as clarified in [hyperref][https://openreview.net/forum?id=sREOiWynXf&noteId=lSfPPiWIvP].
>
> > Different metrics are used ... makes it difficult to compare across the benchmarks
>
> **Response:** The choice of metrics is **intentional** and **benchmark-specific**: each metric is selected to be compatible with the **design** of the dataset and to **faithfully capture information leakage** in that setting.
>
> There is **no single metric** that is universally appropriate across all benchmarks [3]. For example, ROUGE-L can be unreliable for long or free-form generations like for WMDP dataset [4], and accuracy-based metrics are not meaningful for MUSE benchmark [5].
>
> [3] Yuan et al., A closer look at machine unlearning for large language models, 2024.
>
> [4] Li et al., The wmdp benchmark: Measuring and reducing malicious use with unlearning, 2024.
>
> [5] Shi et al., Muse: Machine unlearning six-way evaluation for language models, 2024.
>
> > Explorations are limited to smaller models, on the order of 1-7B parameters ...
>
> **Response:** We agree with the reviewer evaluating larger models would be valuable, but **no unlearned models over 7B** currently exist for TOFU, MUSE, or WMDP [4, 5, 6, 7].
>
> [6] Zhang et al., Negative preference optimization: From catastrophic collapse to effective unlearning, 2024.
>
> [7] Maini et al., Tofu: A task of fictitious unlearning for llms, 2024.
>
> > While I like the RULE approach, it is only applied to TOFU
>
> **Response:** We replied in [hyperref][https://openreview.net/forum?id=sREOiWynXf&noteId=lSfPPiWIvP]
>
> > How does the estimator in Eq. 3 relate to the pass@k estimator derived by Chen et al. (2021) ...
>
> **Response:** Our estimator is related in spirit to the pass@k estimator, but **extends to continuous leakage scores**.
>
> **Pass@k estimator** assumes binary success (pass/fail). Given $n$ samples with $c$ correct ones:
> $$\text{pass@}k=\mathbb{E}\left[1-\frac{\binom{n-c}{k}}{\binom{n}{k}}\right].$$
>
> **Leak@k estimator (Eq. 3)** generalizes this to continuous leakage scores. Let $s(j)\in[0,1]$ denote the sorted leakage scores:
> $$\text{Leak@}k=\sum_{j=1}^{n}\big(s(j)-s(j-1)\big)\left(1-\frac{\binom{j-1}{k}}{\binom{n}{k}}\right).$$
>
> > Across many of the heatmaps, the "retrain" model exhibits significant leakage. Have you thoroughly analyzed what is driving this leakage ...
>
> **Response:** We evaluate a *pre-trained model* that has not been fine-tuned on either the forget or retain sets of the MUSE-News benchmark. As shown in the following table, the pre-trained model already exhibits substantial leakage, with $\widehat{\text{Leak@k}}$ increasing from 0.25 (k=1) to 0.84 (k=128). Therefore, we can conclude that:
> (i) **No benchmark contamination:** leakage appears even without any exposure to the benchmark during fine-tuning.
> (ii) **Not driven by retraining:** retrain inherits leakage rather than introducing it.
> (ii) **Pre-training memorization is the primary source:** the model already encodes the target information prior to any unlearning or retraining.
>
> |Method|Leak@1(↓)|Leak@2(↓)|Leak@4(↓)|Leak@8(↓)|Leak@16(↓)|Leak@32(↓)|Leak@64(↓)|Leak@128(↓)|
> |-|-|-|-|-|-|-|-|-|
> |Pre-trained|0.25|0.37|0.48|0.58|0.66|0.73|0.79|0.84|
>
> > Have you explored RULE on any of the other unlearning benchmarks ... unlearning setting.
>
> **Response:** We replied in [https://openreview.net/forum?id=sREOiWynXf&noteId=lSfPPiWIvP]
>
> > Have you measured the uncertainty in the point estimates ...
>
> **Response:** We repeat the estimation of Leak@k 25 times and report the mean, standard deviation, and 95% confidence interval. The results show that the variance across random seeds **is consistently small**.
>
> |Leak@k|1|2|4|8|16|32|64|128|
> |-|-|-|-|-|-|-|-|-|
> |Mean|0.2610|0.3670|0.4744|0.5733|0.6580|0.7281|0.7848|0.8302|
> |Standard Deviation|0.0011|0.0014|0.0018|0.0021|0.0029|0.0037|0.0045|0.0066|

---

> > ### Author Rebuttal · Reviewer_z5Dj · 2026-04-04
> >
> > Thank you for the rebuttal. I appreciate the additional clarifications, but they do not change my overall assessment or score.
> >
> > Specifically, the rebuttal simply restates the contributions expressed in the paper rather than responding to the specific points in my review. For example, in response to how this paper relates to Chen et al. (2021), the authors state that they "emphasize a key conceptual shift[...]" and then explain that Leak@k works in the opposite direction. This doesn't clarify how the paper contributes beyond Chen et al. nor simply acknowledge that a citation is appropriate. Moving beyond this example, the other responses follow the same pattern, i.e., they re-state what is already stated in the paper. The one area where the authors do provide new evidence is that a pre-trained model never fine-tuned on the benchmark *also scores 0.25–0.84 on Leak@k*. This suggests much of the measured leakage reflects pre-training memorization rather than unlearning failure. The authors do not address this.
> >
> > For these reasons, I cannot increase my original rating.

---

> > > ### Author Response · Authors · 2026-04-06
> > >
> > > >Specifically, the rebuttal ... a citation is appropriate.
> > >
> > > **Response:** We thank the reviewer for engaging in the rebuttal and providing the constructive feedback. In our previous rebuttal, our intention was to provide clarification-centric responses that leverage existing evidence and statements in the paper to address your earlier questions. However, we are happy to further elaborate on these points. We agree that our previous rebuttal did not clearly address the relation to Chen et al. (2021).
> > >
> > > To clarify this connection, **pass@k is the binary special case of Leak@k**. When the underlying metric is binary, i.e., $s_i\in\{0,1\}$, Leak@k reduces exactly to pass@k. Assume there are $c$ correct samples and $n-c$ incorrect ones. Then, the ordered scores satisfy the following property$$s_{(1)}=\cdots=s_{(n-c)}=0, \quad
> > > s_{(n-c+1)}=\cdots=s_{(n)}=1.$$
> > > Substituting into our $\widehat{\text{leak}@k}$ in (3), we yield$$\widehat{\text{leak}@k} = 1-\frac{\binom{n-c}{k}}{\binom{n}{k}},$$
> > > which is exactly the unbiased pass@k estimator, presented in (Chen et al., 2021).
> > >
> > > Beyond this derivation, our contribution is a **generalization to a different statistical object required for unlearning evaluation**:
> > >
> > > First, **pass@k assumes binary correctness**, whereas **Leak@k operates on continuous leakage scores $s(x)\in [0,1]$**. This enables modeling **partial leakage** (e.g., paraphrases, semantic overlap), which pass@k cannot capture.
> > >
> > > Second, pass@k asks whether the model can succeed at least once (best-case capability), whereas Leak@k asks whether the model can fail even once (worst-case safety). In unlearning, a single leaking sample indicates failure, making this formulation necessary.
> > >
> > > We will revise the paper to explicitly acknowledge that Leak@k is inspired by the pass@k estimator in (Chen et al., 2021), and to clearly articulate the differences between the two as discussed above.
> > > >Moving beyond this example, the other ... stated in the paper.
> > >
> > > **Response:** We apologize for the lack of clarity in our previous responses. We complement them with further details.
> > > >While I like the RULE approach, it is only applied to TOFU.
> > >
> > > In response to your concern, we extended the evaluation to the MUSE benchmark using [RULE-NPO](https://openreview.net/forum?id=sREOiWynXf&noteId=lSfPPiWIvP) (our response to 3rd Q. of R. qf1u). The results show that RULE continues to outperform existing unlearning methods, indicating that its effectiveness is **not limited to the synthetic TOFU setting**, but generalizes to a more realistic benchmark.
> > > >There are no estimates of uncertainty...
> > >
> > > We evaluated Leak@k across multiple random seeds and report mean, standard deviation, and confidence intervals in [Table](https://openreview.net/forum?id=sREOiWynXf&noteId=9JfekzBTe0). The shown consistently small variance indicates that the findings are not driven by sampling noise, but show stable differences across methods.
> > > >The empirical pattern that "leakage consistently increases with k" is entirely mechanical... focus on the level of the leakage.
> > >
> > > We introduced a principled decomposition of leakage into: (1) **early-stage leakage** (captured by $\alpha_0$); (2) **decay / flatness under sampling** (captured by $\alpha_1$). Larger $\alpha_0$ means larger **early-stage** leakage; larger $\alpha_1$ means **faster decay** of leakage growth, i.e., stronger saturation/flatter leakage curve. We measured these parameters for various models and presented in [response](https://openreview.net/forum?id=sREOiWynXf&noteId=lSfPPiWIvP) (provided as our response to 1st Q. of R. qf1u).
> > > >The one area where the authors do provide new evidence ... address this.
> > >
> > > **Response:** We agree that the additional experiment shows the sensitive leakage already exists in the pre-trained model. However, the purpose of unlearning is precisely to remove a model’s ability to generate sensitive information that it already memorizes, **regardless of whether that information originates from pre-training or later fine-tuning**. The fact that the starting model memorizes sensitive content is not a counterargument to our evaluation, it is the very reason unlearning is needed in the first place.
> > >
> > > The relevant question, thus, is not simply where the leakage comes from, but whether post-unlearning models successfully remove that leakage. Our evaluation is performed on the **post-unlearning models**, and the main finding is that they still exhibit substantial leakage under probabilistic decoding. e, the central result remains: **existing unlearning methods do not truly remove/suppress that memorized information when the model is evaluated under probabilistic decoding**.
> > >
> > > This also clarifies that Leak@k is not a metric artifact: it does not introduce leakage, but rather **exposes the probability that memorized information re-emerges under repeated sampling**, which is precisely the setting where unlearning should be effective.

---

### Official Review · Reviewer_qf1u · 2026-03-13

**Soundness:** 3
**Presentation:** 2
**Significance:** 3
**Originality:** 3
**Overall Recommendation:** 4
**Confidence:** 5

**Summary:**

This paper studies LLM unlearning under probabilistic decoding rather than greedy decoding. The main claim is that many existing unlearning methods appear successful under deterministic evaluation but still leak forgotten information when multiple stochastic generations are sampled. To quantify this effect, the paper introduces leak@k, a meta-metric defined over the worst leakage among k sampled responses, together with an unbiased estimator. The paper then evaluates several unlearning methods on TOFU, MUSE-News, and WMDP under different decoding settings, and finds that leakage generally increases with the number of sampled generations. Finally, the paper proposes RULE, a generation-augmented unlearning procedure that dynamically expands the forget set using leaked samples, and shows promising results mainly on TOFU.

**Compliance With Llm Reviewing Policy:**

Affirmed.

**Final Justification:**

I raise my score to 4. See Rebuttal Acknowledgement for reasons.

**Key Questions For Authors:**

- What is the precise target of unlearning in this paper? If the retrain model can still output the target answer, why should that behavior be counted as leakage rather than retained-data support or benchmark ambiguity?
- Can RULE be validated beyond TOFU? Since the paper presents RULE as an initial solution, evidence on at least one additional benchmark would make the practical contribution much stronger.
- Why is leak@k the right evaluation objective for unlearning, beyond being an unbiased worst-case multi-sample estimator? In particular, how should one interpret the choice of k in practice, and what is the intended relationship between k and utility?

If the authors address my concers, I am willing to raise my score.

**Limitations:**

No. The paper includes an impact statement, but the limitations discussion is not sufficiently explicit. The paper should more clearly acknowledge: (i) the ambiguity around retrain as the gold-standard upper bound, (ii) the dependence on benchmark-specific core metrics, (iii) the lack of a principled recommendation for choosing decoding settings and k, and (iv) the fact that RULE is only validated on TOFU.

**Strengths And Weaknesses:**

Strengths:
- The paper addresses an important and practically relevant problem. Evaluating unlearning only under greedy decoding can indeed miss leakage that appears under realistic stochastic decoding, so the paper raises a useful caution for the community.
- The idea of evaluating unlearning through a multi-sample leakage lens is interesting, and leak@k is a reasonable way to formalize this concern.
- The empirical study is fairly broad, covering multiple benchmarks, methods, and decoding settings, which makes the central observation non-trivial.

Weaknesses:
- The paper still does not clearly define what should count as successful unlearning when the retrain-from-scratch model itself can sometimes produce the “forgotten” answer. In that case, it is unclear whether leak@k is measuring unlearning failure, benchmark contamination / residual support from the retain distribution, or general alignment/refusal behavior. This weakens the conceptual foundation of the paper.
- The theoretical part mainly shows that the proposed estimator is unbiased, but this does not really justify why leak@k is the right target for unlearning evaluation. In its current form, the theory feels disconnected from the main claim.
- The proposed mitigation, RULE, is only validated on TOFU, and the paper explicitly leaves extension to MUSE and WMDP as future work. As a result, the paper’s positive algorithmic takeaway is still limited to a narrow setting.
- The presentation needs substantial polishing. The manuscript appears over-formatted (e.g., aggressive spacing / \vspace usage), table captions should be placed above tables, and the section budget is unbalanced; for example, the section introducing RULE is too short relative to its importance.
- Related work should better include more recent LLM unlearning papers, e.g.,
    - Rethinking Machine Unlearning for Large Language Models. NMI 2024.
    - LLM Unlearning via Loss Adjustment with Only Forget Data. ICLR 2025.
    - LLM Unlearning with LLM Beliefs. ICLR 2026.

---

> ### Author Rebuttal · Authors · 2026-03-31
>
> > The paper still does not clearly define what should count as successful unlearning...This weakens the conceptual foundation of the paper.
>
> **Response**: Successful unlearning should exhibit consistent behaviors under repeated sampling, rather than being judged by **single-instance leakage**. To this end, we introduce a principled decomposition of leakage into: (1) **early-stage leakage** (captured by $\alpha_0$); (2) **decay/flatness under sampling** (captured by $\alpha_1$). Starting from the estimated leakage curve $L_k:=\widehat{\mathrm{Leak}@k}$ in (3), we define the leakage rate as$$h_k:=\frac{L_{k+1}-L_k}{1-L_k},$$
> where $h_k$ measures the **normalized increase in leakage from one additional sample**, i.e., how much new leakage is exposed at step $k+1$ relative to what has not yet been exposed. In our setting, $h_k$ typically varies with $k$, reflecting the fact that leakage is not governed by a single fixed probability. To summarize how fast $h_k$ decays with $k$, we fit the log-log model $$-\log h_k=\alpha_0+\alpha_1\log k.$$
> Here, $\alpha_0$ is the **intercept**, controlling the initial scale of leakage growth, $\alpha_1$ is the **decay rate**, controlling how quickly the leakage rate decreases as sampling budget increases. Larger $\alpha_0$ means larger **early-stage** leakage; larger $\alpha_1$ means **faster decay** of leakage growth, i.e., stronger saturation/flatter leakage curve.
> Therefore, a successful unlearning method should ideally achieve: (1) **large $\alpha_0$** (**small** early-stage leakage), (2) **large $\alpha_1$**. In the below table, we observe that BLUR-NPO and NPO-SAM achieve the largest $\alpha_0$, indicating lowest initial leakage. Further, BLUR-NPO and Retrain achieve the highest $\alpha_1$, implying leakage saturates fastest.
>
> |Method|$\alpha_0$|$\alpha_1$|
> |-|-|-|
> |BLUR-NPO|**2.05**|**0.73**|
> |GradDiff|1.84|0.69|
> |NPO|1.65|0.63|
> |NPO-SAM|**2.03**|0.58|
> |Original|1.26|0.66|
> |Retrain|1.66|**0.75**|
> |SimNPO|1.49|0.65|
> Table. $\alpha_0$ and $\alpha_1$ values for the MUSE-News benchmark with the LLaMA2-7B model using $(T,p)=(1.0,1.0)$.
>
> > The theoretical part mainly shows that...from the main claim.
>
> **Response**: The goal is not to prevent a single output, but to ensure that **sensitive information cannot be recovered through repeated queries**. This directly motivates to define $L_k$ which measures the probability that the information resurfaces within $k$ attempts. The theory then supports this target in two ways. First, the unbiased estimator ensures that $L_k$ is measured correctly from finite samples. Second, we complete the analysis by defining $h_k$ which characterizes how leakage evolves under probabilistic decoding. Please check our response to your first comment.
>
> > The proposed mitigation, RULE, is only validated on TOFU...limited to a narrow setting.
>
> **Response:** To demonstrate scalability, we extend RULE on **MUSE-News**, a corpus-based dataset. We: (1) Sample 200 paragraphs and use GPT-4o to generate QA pairs; (2) Use BLUR-NPO as base model; (3) Generate a new training set and identify leakage pairs via ROUGE-L on 50 samples for each question; (4) Unlearn the base model and update; (5) Iterate (3) and (4) for 3 rounds. We show that RULE **consistently reduces leakage** across all $k$, outperforming Retrain and BLUR-NPO to achieve **SOTA** Leak@k performance. Training time:~1h.
>
> |Method|Leak@1|Leak@2|Leak@4|Leak@8|Leak@16|Leak@32|Leak@64|Leak@128|
> |-|-|-|-|-|-|-|-|-|
> |Retrain|0.2258|0.3012|0.4480|0.6139|0.6720|0.7604|0.8329|0.8886|
> |BLUR-NPO|0.1237|0.2809|0.3500|0.4961|0.5420|0.6459|0.7389|0.7899|
> |**RULE-NPO**|**0.1228**|**0.2182**|**0.2719**|**0.3462**|**0.3865**|**0.4518**|**0.4972**|**0.5245**|
>
> > The presentation needs substantial polishing.
>
> **Response**: We will edit and polish the format of the paper for the final version.
>
> > Related work should better include more recent LLM unlearning papers, e.g.,...
>
> **Response**: We will cite more recent LLM unlearning works.
>
> > What is the precise target of unlearning in this paper?...
>
> **Response**: Please check our response to your first comment.
>
> > Why is leak@k the right evaluation objective for unlearning, beyond being an unbiased worst-case multi-sample estimator?...
>
> **Response**: We discussed why Leak@k is the right evaluation objective for unlearning in our response to your second comment. Leak@k should be interpreted as a **budgeted risk** metric. It measures the probability that sensitive information appears within $k$ sampled responses. As in pass@k [1], the choice of $k$ is *not provided* but an attempt budget reflecting the application [1]. Also, in safety settings, repeated sampling can significantly amplify risk compared to single-shot evaluation [2]. The objective of a successful unlearning is that the model's Leak@k being **flat and low** regardless of $k$.
>
> [1] Chen et al., Evaluating Large Language Models Trained on Code, 2021.
>
> [2] Hughes et al., Best-of-N Jailbreaking, 2024.

---

> > ### Author Rebuttal · Reviewer_qf1u · 2026-04-01
> >
> > Thank you for the detailed rebuttal. The authors have addressed most of my concerns, in particular by clarifying the motivation and interpretation of the proposed evaluation, and by providing additional evidence for the practical value of the mitigation. While I still think some parts of the paper could be further polished in the final version, the rebuttal has substantially improved my confidence in the work. I therefore decide to raise my score to 4. Good luck!

---

> > > ### Author Response · Authors · 2026-04-04
> > >
> > > Thank you for taking the time to review our paper and for recognizing our efforts. We sincerely appreciate your feedback and support!

---

### Decision · Program_Chairs · 2026-04-30

**Decision:**

Accept (regular)

**Comment:**

This paper adds an important perspective and contributes to the literature on the topic of "LLM unlearning". It makes a convincing case for new measures and methods that estimate unlearning by looking beyond deterministic greedy decoding.